# The good, the bad and the ugly of lockdowns during Covid-19

**Talita Greyling[1]◉, Stephanie Rossouw[2]◉\*, Tamanna Adhikari[1]◉**

**1** School of Economics, University of Johannesburg, Johannesburg, South Africa, **2** School of Economics, Auckland University of Technology, Auckland, New Zealand

◉ These authors contributed equally to this work.
\* stephanie.rossouw@aut.ac.nz

## Abstract

Amidst the rapid global spread of Covid-19, many governments enforced country-wide lockdowns, with likely severe well-being consequences. In this regard, South Africa is an extreme case suffering from low levels of well-being, but at the same time enforcing very strict lockdown regulations. In this study, we analyse the causal effect of a lockdown and consequently, the determinants of happiness during the aforementioned. A difference-in-difference approach is used to make causal inferences on the lockdown effect on happiness, and an OLS estimation investigates the determinants of happiness after lockdown. The results show that the lockdown had a significant and negative impact on happiness. In analysing the determinants of happiness after lockdown, we found that stay-at-home orders have positively impacted happiness during this period. On the other hand, other lockdown regulations such as a ban on alcohol sales, a fear of becoming unemployed and a greater reliance on social media have negative effects, culminating in a net loss in happiness. Interestingly, Covid-19, proxied by new deaths per day, had an inverted U-shape relationship with happiness. Seemingly people were, at the onset of Covid-19 positive and optimistic about the low fatality rates and the high recovery rates. However, as the pandemic progressed, they became more concerned, and this relationship changed and became negative, with peoples' happiness decreasing as the number of new deaths increased.

## 1. Introduction

In an attempt to curb the spread of Covid-19 and minimise the loss of life, governments around the world have imposed their version of mandatory self-isolation through implementing lockdown regulations. Unfortunately, restricting people's mobility and depriving them of what matters most might intensify the negative effect on happiness levels.

In an extreme country case, this might be amplified. In this study, we treat an extreme country as a country with very strict lockdown regulations, with likely high infection rates, amidst low levels of well-being. We define well-being as those aspects of life that society collectively agrees are important for a person's quality of life, happiness and welfare. One of the dimensions of well-being, material (income) hinges on a bleak economic outlook.

To this end, our primary aim in this study is to use the Gross National Happiness Index (GNH), a real-time measure of well-being, derived from Big Data, to investigate if lockdown

of the research. 1. Prof T Greyling: University of
Johannesburg via the University Research Fund. 2.
Dr Stephanie Rossouw: Auckland University of
Technology via the Faculty of Business, Economics
and Law. The funders had no role in study design,
data collection and analysis, decision to publish, or
preparation of the manuscript.

**Competing interests:** The authors have declared
that no competing interests exist.

regulations in itself caused a decrease in happiness. Secondly, we determine which factors matter most (factors significantly influencing happiness) to happiness under these changed circumstances. We accomplish these aims by using two econometric techniques: difference-in-difference (DiD) and ordinary least squares (OLS).

Against this backdrop, the current study makes several contributions to the literature:

i.   Determining whether *lockdown regulations cause a decline in happiness*–in an *extreme country* case scenario.

ii.  Investigating specifically the *determinants of happiness* during a lockdown, whereas other studies have focused on mental well-being and related matters (see section 2).

iii. Being one of the few studies (see also Rossouw, Greyling and Adhikari; Greyling, Rossouw and Adhikari [1, 2]) that investigates the effect of lockdown on happiness making use of real-time Big *Data*. Other studies such as Hamermesh [3] and Brodeur et al. [4] also use Big Data, though limited to *Google Trends* (see section 2).

These results give policymakers the necessary information to take action in increasing the happiness of the nation and set the scene for increased economic, social and political well-being. It also allows them to reflect on happiness outcomes due to their policy actions. An additional benefit of the current study is that policymakers do not need to wait for extended periods to see the consequences of their policies, as we are making use of real-time data, with immediate information. Usually, policymakers can only evaluate their own decision making, with significant time-lags, prolonging the implementation of corrective actions.

Our results indicate lockdown itself causes a decrease in happiness. Furthermore, in an *extreme country case* (a country under stringent lockdown regulations coupled with low levels of well-being) what matters most to happiness under lockdown is the factors directly linked to the regulations that were implemented. These factors can be classified as (i) social capital issues; lack of access to alcohol (and tobacco), increased social media usage, and more time to spend at home, of which all are negatively related to happiness except the stay-at-home factor, and (ii) economic issues; concerns over jobs and the threat of retrenchments, which are negatively related to happiness. The finding on the stay-at-home order is interesting as even though lockdown itself caused a decline in happiness, it seems that people adjust and over time begin to appreciate the benefits of staying at home.

Noteworthy is that Covid-19, proxied by new deaths per day, had an inverted U-shape relationship to happiness. Seemingly people were, at the onset of Covid-19, positive and optimistic as the fatality rate was relatively low and recovery rates high. However, as the pandemic progressed, they became more concerned, and this relationship changed and became negative, with peoples' happiness decreasing as the number of new Covid-19 deaths increased.

The rest of the paper is structured as follows. The next section contains a brief background on South Africa and briefly discusses literature about happiness and studies conducted on the impacts of pandemics and consequently lockdown regulations. Section 3 describes the data, the selected variables and outlines the methodology used. The results follow in section 4, while the paper concludes in section 5.

## 2. Background and literature review

### 2.1 South Africa

In this study, we focus on South Africa because it presents us with a unique case to investigate the effect of a lockdown on happiness when levels of well-being are already low. Health and income, two dimensions of well-being, was significantly affected, although in opposite

directions. Health was positively affected by the lockdown since it limited the spread of Covid-19. At the stage of writing the paper (3 June 2020), the number of new Covid-19 cases were nearly 120,000 (John Hopkins University [5]). On the other hand, the economic outlook of the country, and therefore peoples' incomes, was negatively affected. This opposite effect has led to significant debates on the value of the implementation of the lockdown.

Furthermore, South Africa implemented one of the most stringent lockdown regulations (comparable to the Philippines and Jordan), which exacerbated the costs to well-being and the economy while already experiencing a severe economic downturn. Therefore, South Africa is an example of an *extreme country case* which unfortunately amplifies the effects of the difficult choices made by policymakers. Therefore, we take advantage of this unique country case and determine how stringent lockdown regulations impact happiness during a one in 100-year event.

In South Africa, there are five levels of differing lockdown regulations, with alert level 5 being the most stringent and alert level 1 the most relaxed. The idea behind these levels is to curb the spread of Covid-19 and give time to South Africa's health system to prepare itself. Additionally, as they move down in levels, South Africans receive increasingly more of their previous liberties back. During level 5, which was announced 23 March 2020 and implemented on 27 March 2020, South Africans were only allowed to leave their homes to purchase or produce essential goods. All South Africans were instructed to work from home, there was no travel allowed, the sale of alcohol and tobacco were banned, people were not allowed to exercise outside their homes, and the police and defence force ensured compliance to the restrictions. South Africa moved to level 4 lockdown on 01 May 2020. With this move, they received back the ability to exercise outside from 6 am—9 am, purchase more than just essential goods, including food deliveries as long as it was within curfew.

Interestingly, the sale of alcohol and tobacco was still banned. On 01 June they moved to level 3, allowing restricted sales of alcohol (Mondays to Thursdays) and the re-opening of certain businesses. However, the services industry, especially beauty and tourism, remained closed. At the time of writing this paper, South Africa was still under level 3 lockdown.

Whereas everybody understands that the Covid-19 infections curve needs to be flattened, there are grave concerns that these stringent lockdown regulations will also flatline South Africa's well-being and economy. Before the Covid-19 lockdown, South Africa's average happiness levels were 6.32 compared to an average of 7.23 and 7.16 in Australia and New Zealand, respectively (Greyling et al. [2]). Furthermore, South Africa had a 29 per cent unemployment rate, and the gross domestic product (GDP) has been estimated to shrink by 7 per cent in 2020 (Bureau of Economic Analysis [6]). According to the South African Reserve Bank [7], an additional 3 to 7 million people can potentially become unemployed as a direct consequence of the pandemic, thereby increasing unemployment rates to approximately 50 per cent. The country's sovereign credit rating was downgraded to junk status in March 2020, which impacted on political stability, the level of the national debt and debt interest payments. Add to this already grim situation, the fact that consumption of South Africans has been declining in 2020, with a significant decrease seen after lockdown, then one can very easily see how the well-being and happiness levels in South Africa can plummet.

## 2.2 Happiness

Why should we care whether people's happiness is adversely impacted by not only a global pandemic but also by the response from the government? The studies of Helliwell, Layard, Stiglitz et al., Veenhoven, Diener and Seligman and others [8–12], have shown beyond a shadow of a doubt that if policymakers want to maximise the quality of life of their citizens,

they need to consider subjective measures of well-being. Piekalkiewicz [13] states that happiness may act as a determinant of economic outcomes: it increases productivity, predicts one's future income and affects labour market performance. By measuring happiness, individuals themselves reveal their preference and assigned priority to various domains, which cannot be identified by a measure such as GDP. As was pointed out by Layard [9], while economists use exactly the right framework for thinking about public policy, the accounts we use of what makes people happy are wrong. In layman's terms, we say that utility increases with the opportunities for voluntary exchange. However, Layard [14] argues that this overlooks the significance of involuntary interactions between people. Policymakers should formulate policy to maximise happiness or well-being, as is the main aim of many constitutions. This can be achieved by directing economic, social, political and environmental policy to maximise well-being while acknowledging that people's norms, aspirations, feelings and emotions are important. Thereby underscoring that understanding and measuring happiness should be an integral part of the efforts to maximise the quality of life.

On the other hand, if people's happiness is negatively affected, such as it was in the wake of the Covid-19 pandemic and the implementation of lockdown regulations, there are far-reaching consequences.

These consequences are as follows:

i. Social capital: unhappier people display less altruistic behaviour in the long run (Dunn et al. [15]). They are also less active, less creative, poor problem solvers, less social, and display more anti-social behaviour (Lyubomirsky et al. [16]). If unhappier people display more anti-social behaviour, South Africa could see an increase in behaviour such as unrests, violent strikes and perhaps higher crime rates.

ii. Health care: unhappier people are less physically healthy and die sooner (Lyubomirsky et al. [16]). Additionally, unhappy people engage in riskier behaviour such as smoking and drinking, thereby placing unnecessary pressure on national health systems.

iii. Economic: unhappy workers are typically less productive, in particular in jobs that require sociability and problem solving (Bryson et al. [17]). If an economy can raise the rate of growth of productivity, by ensuring their workers are happier, then the trend growth of national output can pick up.

## 2.3 Literature on the determinants of happiness during a lockdown

Having established that policymakers should strive to maximise the happiness of their people, it is necessary to know what determines happiness. Previous studies have investigated, at a macro-level, what influences happiness and found that economic growth, unemployment and inflation play a significant role (Stevenson and Wolfers, Perović, Sacks et al. [18–20]). However, these studies were conducted during 'normal' periods and not under such conditions that are currently plaguing the world. The current study will have the opportunity to investigate this exact question, namely what determines happiness during a lockdown driven by a global health pandemic.

Naturally, the number of studies being conducted to examine the effect of Covid-19 is growing exponentially. This increasing interest in the effect of a global pandemic as well as the policies implemented by governments on peoples' well-being, come on the back of relatively few studies conducted during prior pandemics such as SARS and the H1N1. When SARS hit in 2002 and then again when H1N1 hit in 2009, scholars were only truly starting to understand that for governments to formulate policies to increase well-being, you needed to measure well-

being. Of the current studies being conducted on the effect of Covid-19 or lockdown regulations on all affected domains, not many studies are in a position to use real-time Big Data, such as we do.

In layman's terms, Big Data is a phrase used to describe a massive volume of both structured (for example stock information) and unstructured data (for example tweets) generated through information and communication technologies such as the Internet (Rossouw and Greyling [1]). At the time of writing this paper, the following studies were closest aligned with our study and focused on:

i.  nationwide lockdown on institutional trust, attitudes to government, health and well-being, using survey data collected at two points in time (December 2019 and April 2020) (1003 respondents) (Sibley et al. [21]). Their preliminary results showed a small increase in people's sense of community and trust. However, they also found an increase in anxiety/depression post-lockdown and hinted at longer-term challenges to mental health.

ii.  the happiness of married and single people while in government-imposed lockdown by running simulations to formulate predictions, using Google Trends data (Hamermesh [3]). Not surprisingly, married people were more satisfied with life than single people.

iii.  the timing of decision-making by politicians to release lockdown based on a comparison of economic benefits with the social and psychological benefits versus the cost, increase in deaths if policymakers released lockdown too early (Layard et al. [22])

iv.  the stages of GNH using a Markov switching model in New Zealand (Rossouw et al. [23]). They found that happiness was at a lower level and the unhappy state lasted longer than was expected. Furthermore, they found that the factors important for New Zealand's happiness post-Covid-19 were related to international travel, employment and mobility.

v.  exploring Covid-19 related determinants of life dissatisfaction and feelings of anxiety in a cross-country study using survey data collected between 23 March and 30 April (de Pedraza et al. [24]). They found that persons with poorer general health, without employment, living without a partner, not exercising daily and those actively seeking out loneliness report higher dissatisfaction and higher anxiety. Additionally, they found that the effect of Covid-19 on dissatisfaction and anxiety levels off with a higher number of cases.

## 2.4 Literature on the causal effect of a lockdown

To the knowledge of the authors, there are only two papers that investigated the causal effect between lockdowns and population well-being. Brodeur et al. [4] investigated the changes in well-being (and mental health) in the United States and Europe after a lockdown was implemented, using Google Trends data. They found an increase in searches for loneliness, worry and sadness, which indicated a negative effect on mental health. Greyling et al. [2] conducted a cross-country study investigating the effect of lockdown on happiness. They found that lockdown caused a negative effect on happiness, notwithstanding the different characteristics of the countries (South Africa, New Zealand and Australia), the duration and the type of lockdown regulations. When they compared the effect size of the lockdown regulations, they found that South Africa, with the most stringent lockdown regulations incurred the greatest happiness costs.

Brodeur et al. [4] study analysed data from one Big Data source, Google Trends and collected data for a short period between only 01 January 2019 and 10 April 2020 in countries that had introduced a full lockdown by the end of this period. Greyling et al. [2] study used

both Google Trends and the GNH index but did not investigate the determinants of happiness after lockdown for an extreme country case.

In summary, taking all of the above into consideration, there is not one study which determines *causality between lockdown and happiness and analyses the determinants of happiness in an extreme country case* using *real-time*, *Big Data*. Therefore, our study is the first of its kind.

## 3. Data and methodology

### 3.1 Data

To estimate the causal effects of a lockdown on happiness, we use a Difference-in-Difference (DiD) approach (see section 3.3.1). The technique compares happiness (dependent variable), before and after the treatment (the lockdown) to a counterfactual time period in the year before. For the control period, we select the same time period, with the same number of days in 2019, corresponding to the number of days in 2020, thus 152 days in each year (01 January 2020 to 03 June 2020, excluding 29 February 2020). Our results should thus be interpreted as the average impact of the lockdown on happiness, comparing pre and post-lockdown in 2020 to the same time period in 2019, which we assume had normal levels of Gross National Happiness (see a discussion on the GNH in section 3.2.1). In this manner, we also account for seasonal trends in happiness.

In the analyses, we make use of daily data for South Africa. As high-frequency data available at almost real-time, is scarce, we make use of novel Big Data methodologies to harvest data. Additionally, we use the Oxford Stringency dataset that was released in May 2020, which includes data related to lockdown regulations, such as time-series data on the stay-at-home index, Covid-19 cases and Covid-19 deaths (Hale et al., Roser et al. [25, 26]).

### 3.2 Selection of variables

The selection of the variables included in our estimations is based on the reviewed literature, the contents of tweets related to the lockdown and data availability.

**3.2.1 Gross National Happiness Index–the dependent variable.**   To measure happiness (the dependent variable), we make use of the Gross National Happiness Index (GNH), which was launched in April 2019 (Greyling, Rossouw and Afstereo [27]). This project measures the happiness (mood) of a country's citizens during different economic, social and political events.

Since February 2020, the researchers extended the project that initially analysed the sentiment of tweets, to incorporate the analysis of the emotions underpinning tweets. The team did this to determine which emotions are most prominent on specific days or events.

To construct the GNH index, the researchers use Big Data methods and extract tweets from the voluntary information-sharing social media platform Twitter. Big Data, such as Twitter, provides real-time information for policymakers to assist them when facing short-term deadlines with imperfect information. Big Data also allows governments to 'listen' and capture those variables which their citizens deem to be important for their well-being, rather than relying on pre-defined economic utility theories. Big Data offers governments the opportunity to observe people's behaviour and not just their opinions. This approach of revealed preferences unveils a reflexive picture of society because it allows the main concerns of citizens (and the priority ranking of those concerns) to emerge spontaneously, and it complements as such the information captured by gross domestic product. Lastly, Big Data does not suffer from non-response bias (Callegaro and Yang [28]).

Greyling, Rossouw and Afstereo [27] apply sentiment analysis to a live Twitter-feed and label every tweet as having either a positive, neutral or negative sentiment. This sentiment classification is then applied to a sentiment-balance algorithm to derive a happiness score. The

happiness scores range between 0 and 10, with five being neutral, thus neither happy nor unhappy.

All tweets per day are extracted, and a happiness score per hour is calculated. The index is available live on the GNH website (Greyling, Rossouw and Afstereo [27]). In South Africa, the average number of tweets extracted for 2020 is 68,524 per day. South Africa has approximately 11 million Twitter users, representing almost 18 per cent of the population (Omnicore [29]). Although the number of tweets is extensive and represents significant proportions of the populations of the countries, it is not representative. However, Twitter accommodates individuals, groups of individuals, organisations and media outlets, representing a kind of disaggregated sample, thus giving access to the moods of a vast blend of Twitter users, not found in survey data.

Furthermore, purely based on the vast numbers of the tweets, it seems that the GNH index gives a remarkably robust reflection of the evaluative mood of a nation. Also, we correlate the GNH index with 'depression' and 'anxiety', derived from the *'Global behaviors and perceptions at the onset of the Covid-19 Pandemic data'* survey, for the period from 01 March 2020 (OFS [30]). We find it negative and statistically significant related, therefore, it seems that the GNH index derived from Big Data gives similar trends to survey data. (We would have appreciated the opportunity to correlate the GNH to a happiness measure–but a happiness measure, as such, was not included in the survey).

Considering the GNH index over time, we found that the index accurately reflects a nation's emotions, for example, when South Africa won the Rugby World Cup on 02 November 2019, the happiness index accurately depicted the joy experienced by South Africans (Fig 1). The hourly happiness score was 7.9 at 13:00, the highest score ever measured, at the exact time that the final whistle was blown to announce the victory of the Springboks over England.

Also, when the famous American basketball player, Kobe Bryant and his daughter Gigi, tragically passed away on 27 January 2020, the happiness index once again captured the

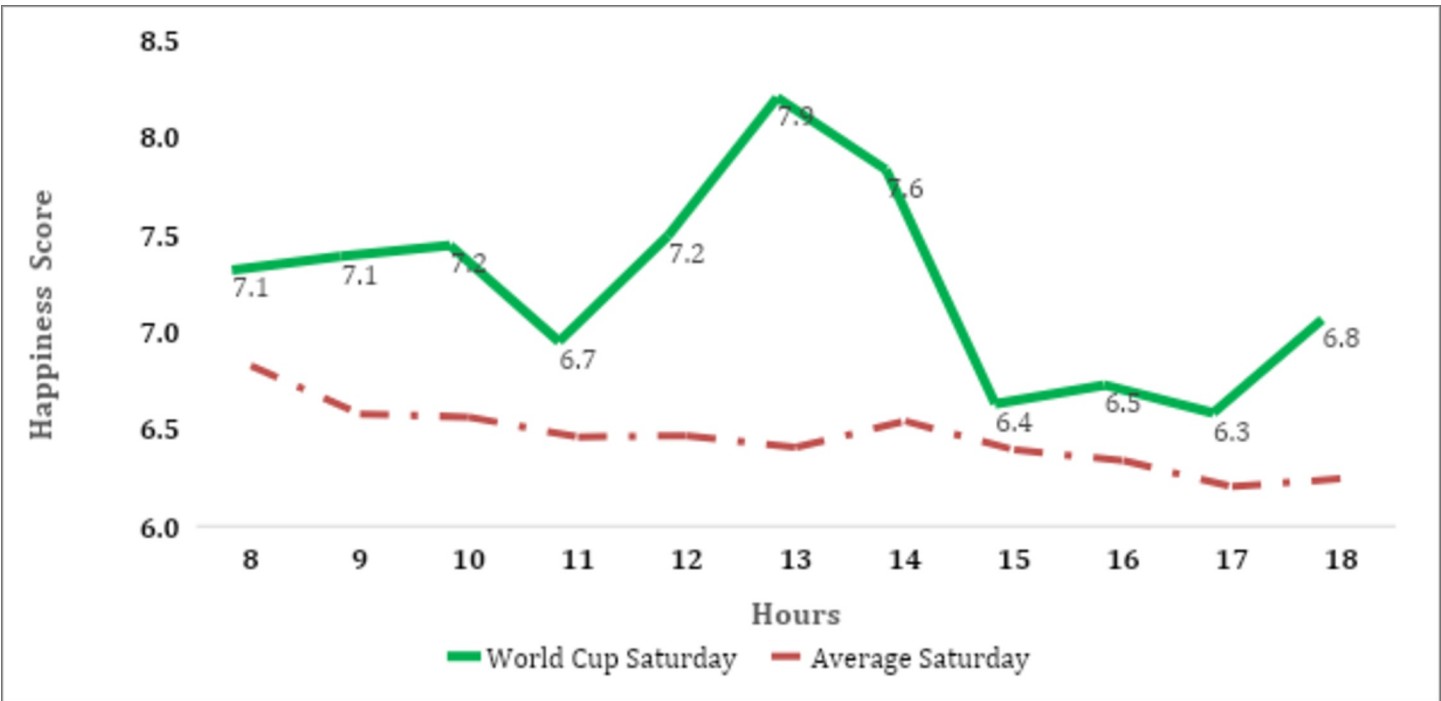

**Fig 1. Hourly happiness levels of South Africans during the Rugby World Cup final match.** Source: Authors' calculations using GNH dataset (Greyling et al. [27]).

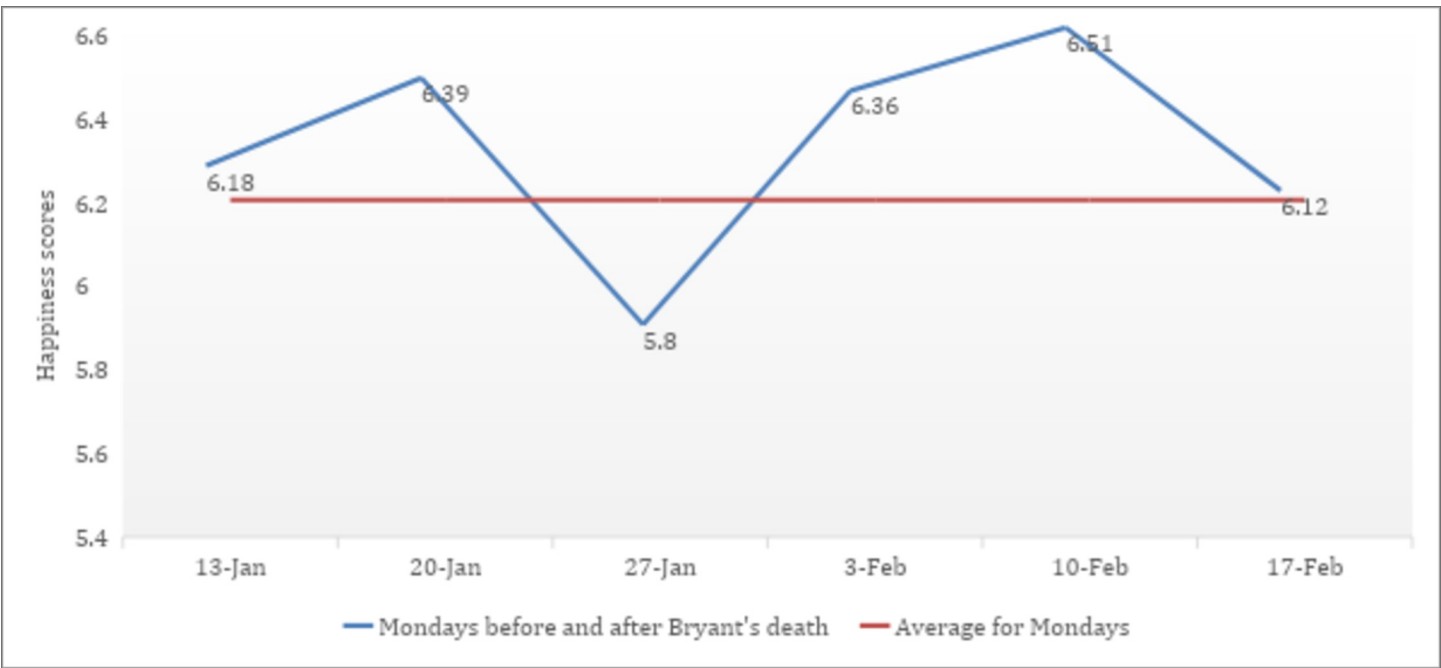

**Fig 2. Happiness level, Kobe Bryant's death.** Source: Authors' calculations using GNH dataset (Greyling et al. [27]).

negative mood of the nation, and the happiness score decreased to 5.8, significantly below the mean (see Fig 2). The result of the GNH mirrors the one determined by the Hedonometer, which recorded an average happiness score of 5.89 on the day of Bryant's death. The top three words that made this day sadder than the previous seven were 'crash', 'died' and 'rip'.

**3.2.2 The selection of covariates included in estimations.** We found ourselves in uncharted territory, as there are limited studies estimating happiness functions during a lockdown (see Brodeur et al., Greyling et al., Rossouw et al. [2, 4, 23]). As a result, we considered these studies and the tweets to determine the factors to consider, which might influence happiness *during a lockdown*, as well as the most tweeted subjects. It was evident from the tweets that the main topics of discussion related to economic concerns, the prohibition of the sale of alcohol and tobacco, the stay-at-home orders and the Covid-19 pandemic itself.

To estimate our difference-in-difference model, we restricted our covariates to the lockdown variable, a year effect, the difference-in-difference estimator and controlled for new Covid-19 deaths, job searches and searches for alcohol. We were restricted in the number of covariates due to the limited observations and potentially encountering the issue of over-identification of the models. Therefore, we selected those variables which were available for both 2019 and 2020, and which were also trending subjects during the lockdown period. We were not able to add a stay-at-home variable which captures the lack of mobility, as the counterfactual time period is then not comparable to 2020.

Lockdown, our treatment variable, divides the sample into two distinct time periods: *before* the announcement of the lockdown, 23 March 2020 and thereafter. We make use of the date of the announcement of the lockdown rather than the date of the implementation, as this showed the severest effect on happiness (see Brodeur et al. [4]).

The Covid-19 pandemic and consequent spread of the virus is the reason for the lockdown. As such, we include the number of new daily Covid-19 deaths as well as its square. This will allow us to control for the likelihood of a U-shaped relationship between the number of

Covid-19 deaths and happiness. Furthermore, there is likely a lagged effect on happiness due to Covid-19 deaths being reported in the media only the following day. Therefore, we lag these variables by one day. We derive the data from the Oxford Stringency data set (Hale et al. and Roser et al. [25, 26]).

To measure jobs (a proxy for future job uncertainty) and the sale of alcohol and tobacco, we use the methodology as set out by Nuti et al. and Brodeur et al. [4, 31] and use daily searches on Google Trends (see also Simionescu and Zimmermann [7]). We considered searches for both the alcohol and tobacco topic; however, the variables follow the same trends during the lockdown period and are highly correlated (r = 0.83). We are, furthermore, restricted in the number of covariates to include in the model and decided to include only 'alcohol' in the estimations. We justify this decision since the ban of alcohol affects a larger proportion of the population. It is estimated that 41 per cent of males and 17.1 per cent of females consume on average 9.3 litres of alcohol per capita annually whereas only 17.6 per cent smoke (Peltzer et al. and Reddy et al. [32, 33]). However, as a robustness check, we also run all estimations using the searches for tobacco.

It should be noted that when we use Google Trends data, it is expressed as an index between 0 and 100 with 0 being the "least" interest and 100 being the "most" interest shown in the topic for the year. However, the series are not comparable across years as the underlying data is sourced from different search requests for each of the two years. To address this, we use a scaling procedure outlined in Brodeur et al. [4]. First, we generate "weekly" interest weights for each day by expressing the average weekly score that a particular daily score fell on, as a proportion of the average yearly score. Then, we multiply the daily scores with these weights to obtain weighted search trends. Finally, we normalise these weighted search trends to render us a score between 0 and 100, which is comparable across years.

Other topics that are trending are related to the 'stay-at-home' orders. The Oxford Stringency dataset includes a time series variable on the stay-at-home orders. It differs on a day to day basis according to its stringency. It is an ordinal variable plus binary of geographic scope. It takes the value 0 if there are no stay-at-home orders and 1 if the government recommends not leaving your house. Value 2 represents people not leaving their homes with the exceptions of daily exercise, grocery shopping, and 'essential' trips. Not leaving your home with minimal exceptions (e.g. allowed to leave only once a week, or only one person can leave at a time, etc.) takes the value 3 (Hale et al. [25]).

Furthermore, we include the number of tweets per day, as it forms part of the Twitter data extracted daily for South Africa (Greyling et al. [27]), which is a proxy for connectivity. It also gauges the opportunity cost of not being able to have face to face interactions, which seems to be negatively related to happiness (Chae and Wilson et al. [34, 35]). Interestingly the number of tweets increased markedly during the lockdown period, from an average of 60,708 to 80,000 tweets per day. Table 1 provides descriptive statistics for the variables included in the estimations.

### 3.3 Methodology

**3.3.1 Difference-in-Difference.** To investigate the causal effect of the lockdown on happiness, as mentioned in section 3.1, we use a Difference-in-Difference estimator (DiD) which compares the GNH for pre and post-lockdown periods in 2020 to a time period assumed to have normal happiness levels in 2019. Specifically, we estimate the following equation:

$$GNH_{i,t} = \alpha_0 + \alpha_1 lockdown_i * Year + +\alpha_2 NewDailyDeaths_{i,t-1} + \alpha_3 NewDailyDeaths^2_{i,t-1} lockdown_{i,t} + \mu_i$$
$$+ \sigma_i + \epsilon_{i,t} \tag{1}$$

**Table 1. Descriptive statistics of the variables included in the estimations of happiness.**

| Variable | 2019 | | | | 2020 | | | |
|---|---|---|---|---|---|---|---|---|
| | Mean | Standard deviation | Min | Max | Mean | Standard deviation | Min | Max |
| GNH | 6.37 | 0.214 | 4.99 | 6.67 | 6.27 | 0.201 | 5.58 | 6.72 |
| New Daily Deaths per million | 0.00 | 0.000 | 0.00 | 0.00 | 1.16 | 2.32 | 0.00 | 10.82 |
| Jobs Searches | 68.57 | 14.04 | 41.57 | 100 | 70.71 | 13.04 | 41.82 | 100 |
| Alcohol | 40.96 | 12 | 1 | 69 | 32.10 | 12.34 | 1 | 57 |
| Stay at Home Index | | | | | 0.901 | 0.911 | 0 | 2 |
| Total Tweets, Logged | 10.92 | 0.288 | 7.65 | 11.25 | 11.06 | 0.23 | 9.5 | 11.55 |

Source: Author's calculations Google trends, Oxford Stringency Index and GNH.

Where $GNH_{it}$ is the daily happiness for South Africa at time $t$. The treatment variable lockdown takes the value of 0 pre-announcement day (23 March) and one post-announcement of lockdown in both the year of the actual lockdown (2020) as well as the year before the lockdown (2019). *Year* is a dummy variable where 1 is the year 2020. We control for new deaths per million with a one-day lag as well as the quadratic effect of new deaths per million on GNH. Additionally, we control for the effect of job and alcohol searches. As a robustness test, we use the number of new Covid-19 cases instead of new Covid-19 deaths (see Table 4 in S1 Appendix).

Our main coefficient of interest is the interaction between the lockdown and the year variable. If it is found to be significant, it provides evidence of a causal effect of the lockdown on the dependent variable, in the current year, notwithstanding the trend in 2019.

**3.3.2 OLS regression.** To determine which factors, matter to happiness after the lockdown, we estimate the following equation:

$$y_t = \alpha_0 + \alpha_1 X_t + \mu_t \tag{2}$$

Here, $y_t$ refers to the Gross National Happiness Index (GNH) for each day and $X_t$ is a vector of several relevant covariates to account for the changes in the happiness levels during the lockdown period. $\mu_t$ is the error term.

Due to the various factors that affect happiness, some of our independent variables may be correlated with the error term, leading to endogeneity concerns. Depending on the direction of the correlation between the error term and the X-variable, the coefficient could be biased upwards or downwards. For instance, the coefficient on the indicator for jobs is likely biased upwards as it, in all likelihood, shows the effect of concerns about jobs as well as some other negative economic shock on happiness. In the absence of panel data or an appropriate instrument, it is difficult to ascertain causality to Eq (2). However, simply correlating the covariates and the error term we find all levels of correlation to be less than 0.3, although a very basic test, this still indicates that the likelihood that endogeneity might bias estimations is relatively small. A natural extension of the work, as better data becomes available with time, would be to address these concerns.

We cannot rule out the probability of autocorrelation and heterogeneity in our data, especially due to its time-series nature. We use robust standard errors to account for this. The choice of our controls, however, comes with a caveat. Seeing as we only have 81 daily observations using a larger battery of covariates would lead to problems arising due to overfitting of the model. This issue is considered in Green [36], who suggests a minimum of 50 observations for any regressions as well as an additional eight observations per additional term. As a robustness test, we included tobacco rather than alcohol products (see Table 5 in S1 Appendix).

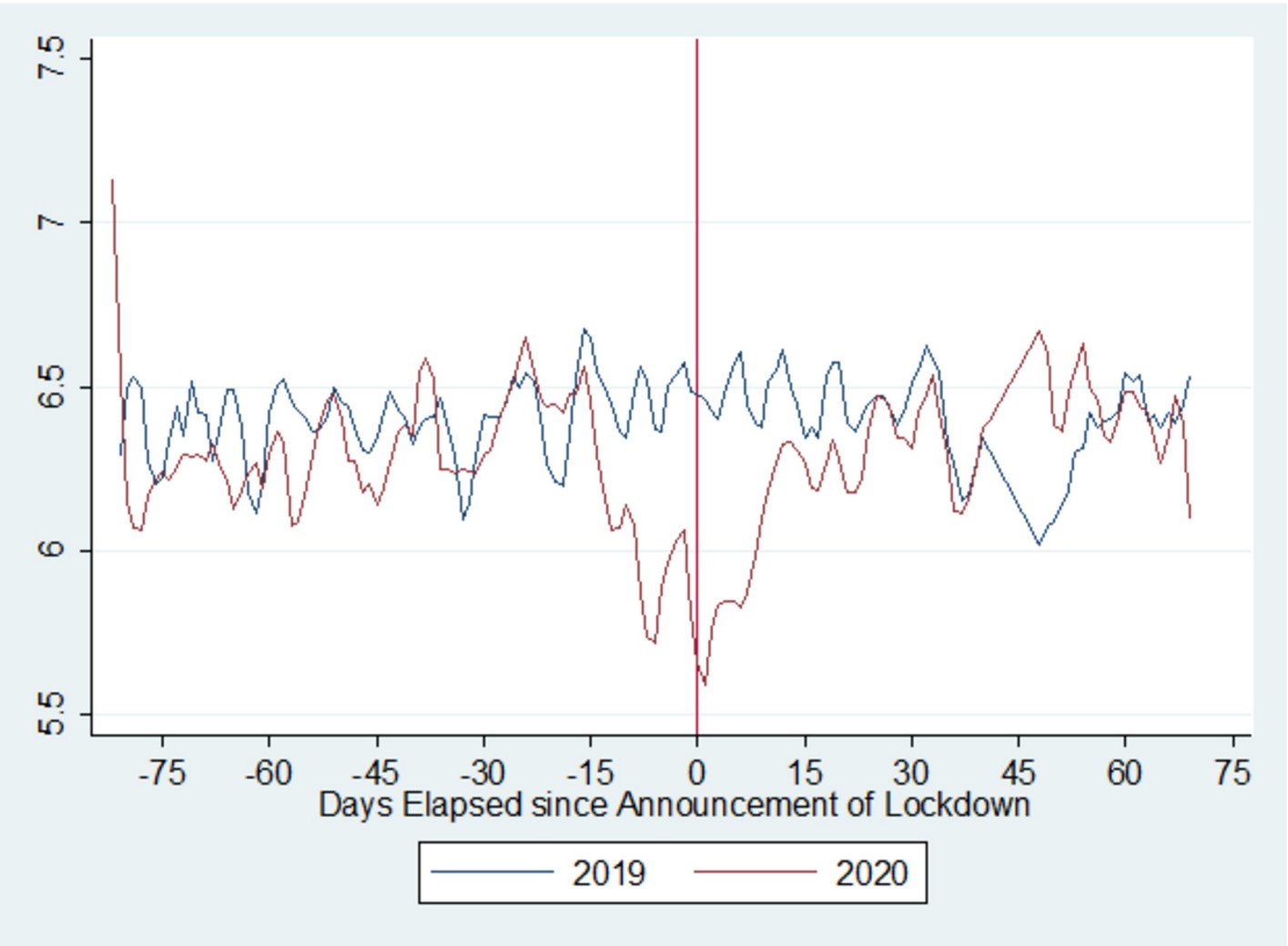

**Fig 3. GNH before and after the lockdown date in 2019 and 2020.** Source: Authors' calculations using GNH dataset (Greyling et al. [27]).

## 4. Results and analysis

### 4.1 Difference-in-Difference estimation

Fig 3 tracks the dependent variable (GNH) over the time period before and after the date of the lockdown (23 March) in the year of the lockdown (2020) and the year preceding it. On the day of the announcement of the lockdown and for a few successive days, we see a sharp downwards departure from the 2019 trend, assumed to be normal.

Table 2 provides the results for the difference-in-difference specification, which helps us to make causal inferences on the effect of the lockdown on the GNH. At the outset, we notice a negative and significant effect of the 'year' variable (p<0.001), showing that on average the GNH was lower in 2020 than in 2019.

We control for trends in job searches (a proxy for job uncertainty) and alcohol searches (a proxy for increased interest in alcohol-related topics in the specification). Both variables show a negative association with GNH, implying if there are more searches for jobs or alcohol, reflecting a scarcity in these items, GNH decreases. The negative effect of alcohol is statistically

**Table 2. Difference-in-Difference estimation.**

| | (1) | |
|---|---|---|
| Dependent Variable: GNH | Coefficient | S.E. |
| Lockdown*Year | -0.101* | (0.055) |
| Year | -0.173*** | (0.027) |
| Job Searches | -0.001 | (0.0008) |
| Alcohol | -0.003*** | (0.0008) |
| Lagged Covid-19 Deaths per million | 0.166*** | (0.023) |
| Lagged Covid-19 Deaths per million Squared | -0.016*** | (0.002) |
| Lockdown F.E. | Yes | |
| Constant | 6.60*** | (0.05) |
| N | 304 | |
| Adjusted $R^2$ | 0.26 | |

Robust Standard errors in parentheses Alcohol represents "lack of beer"

* $p < 0.10$

** $p < 0.05$

*** $p < 0.01$

significant at the 1% level (p<0.001). We also control for lagged new Covid-19 related deaths and lagged new Covid-19 related deaths squared, both are significant (p<0.001). Interesting is the finding of the significant inverted U-shaped relationship between new Covid-19 deaths and happiness (Fig 4). In the earlier stages of the pandemic, with very few new Covid-19 deaths, it appears that people were positive and optimistic as the fatality rates were very low and the recovery rates very high. However, as time progressed, the higher fatality rates turned the relationship around so that the number of new Covid-19 cases were negatively related to happiness.

To determine if the decrease in GNH was due to the lockdown (the treatment) specifically and not just due to the year trend, we must consider the estimated coefficient of the interaction variable 'lockdown and year'. We report a negative and statistically significant coefficient (p-value 0.064) on the interaction variable, indicating that 'lockdown' caused, on average a decrease in GNH of 0.101 points when compared to its mean values for average 2019 values, controlling for the general trend in the two years. Thus, we can conclude that the lockdown caused a decline in GNH in 2020 compared to 2019. The decline of 0.101 may seem small at first but given the low general levels of happiness in South Africa compared to other countries (Greyling et al. [27]) the reduction is substantial.

## 4.2 Regression analysis

To address the second research question, namely, to determine the factors that are related to happiness after the lockdown was implemented, we consider the results of Table 3.

Table 3 shows that job searches (p-value 0.09), searches for alcohol (p-value<0.001) and the number of tweets is negatively related to happiness. In contrast, the stay-at-home index is positively related to happiness (p-value<0.001). The squared relationship between new Covid-19 deaths and happiness is negative and statistically significant (p-value<0.001), indicating that this relationship changed over time as was highlighted in section 4.1. Suppose we consider the relatively low mortality rate (0.02 per cent of confirmed cases in the early stages) compared to countries such as the USA (3.9 per cent), the U.K. (15.4 per cent) and Spain (9.4 per cent). In that case, it could explain the initial positive relationship between the number of new Covid-19

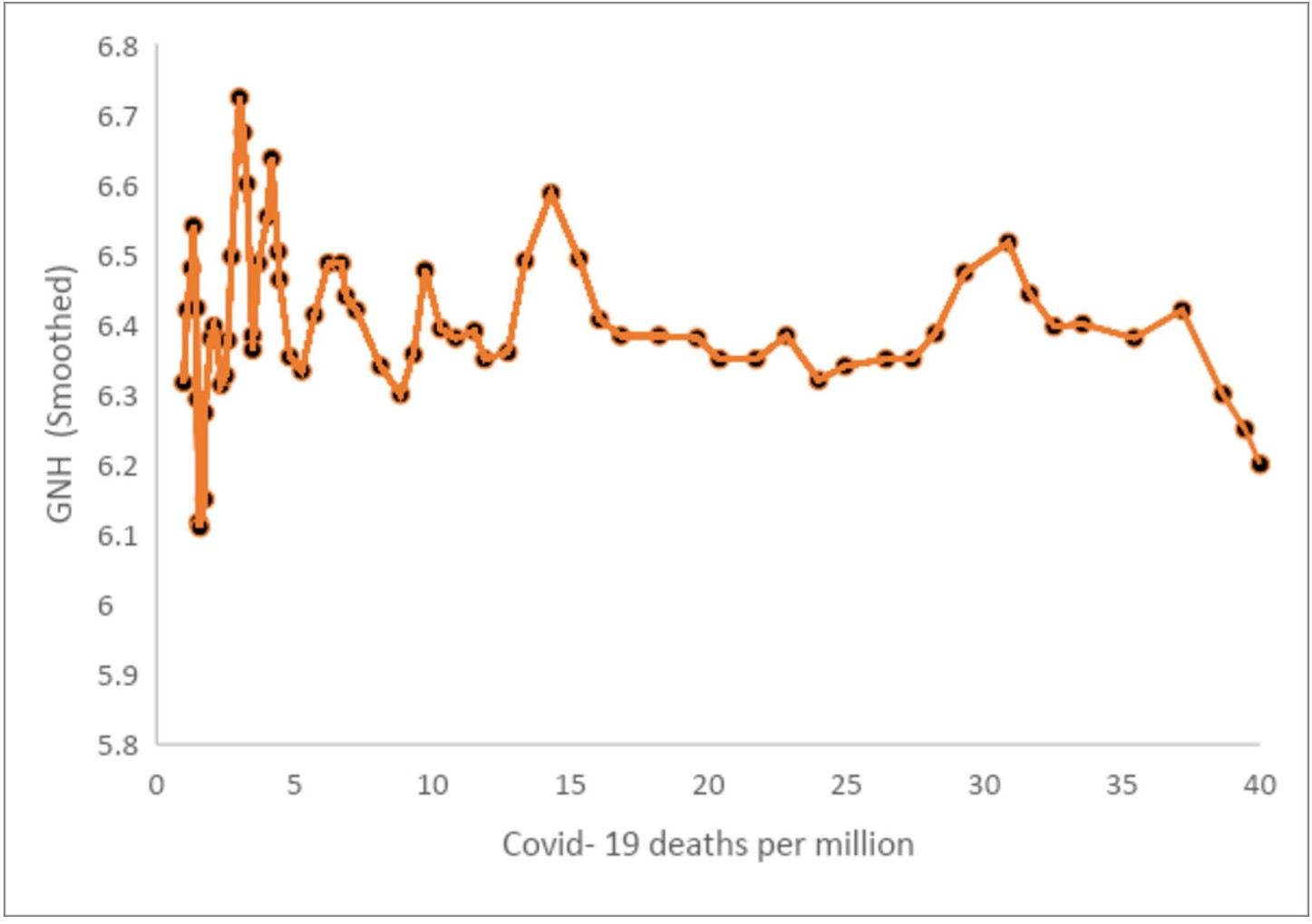

**Fig 4. GNH vs deaths per million.**

**Table 3. Determinants of happiness during the lockdown.**

| Dependent Variable: GNH | Coefficient | S.E. |
|---|---|---|
| Job Searches | -0.0037* | (0.002) |
| Alcohol | -0.0040*** | (0.001) |
| Log Tweets | -0.3682*** | (0.099) |
| Stay at Home Index | 0.2186*** | (0.034) |
| Lagged Covid-19 Deaths per million | 0.0095*** | (0.004) |
| Lagged Covid-19 Deaths per million squared | -0.0002*** | (0.00007) |
| Unhappiest Week F.E. | Yes | Yes |
| Constant | 9.8720*** | (1.172) |
| N | 81 | |
| Adjusted R2 | 0.76 | |

Robust Standard errors in parentheses

\* $p < 0.10$

\*\* $p < 0.05$

\*\*\* $p < 0.01$

deaths and happiness. Although as time passed and the death rate increased (currently, the mortality rate is at 1.5 per cent of confirmed cases), this relationship became negative. As a robustness test, we used the number of new Covid-19 cases and its square instead of the new Covid-19 deaths and found similar results (see Table 4 in S1 Appendix).

As expected, job searches, a proxy for uncertainty about the future job market is negatively related to happiness (p-value<0.001). Analysing the tweets, we realised that this is a major concern, which is closely related to economic concerns. The economic performance of South Africa in the last year has been weak with high levels of unemployment (increase to 50 per cent), low growth rates (GDP is expected to contract with 7 per cent in 2020) and high debt to income ratios (government debt as a percentage of GDP– 80 per cent). In a recent survey conducted by Statistics South Africa on behavioural and health impacts of the Covid-19 pandemic (Statistics South Africa [37]), it was found that 95 per cent of the respondents were very concerned about the economy. In contrast, only 60 per cent was concerned about the Covid-19 pandemic itself. This supports our current findings in that economic factors matter more to South Africans happiness levels than Covid-19 itself.

Alcohol-related searches are also found to be negatively related to happiness (p-value<0.001). Considering the close correlation between alcohol and tobacco products, we can assume that what holds for alcohol products, also holds for tobacco products. As a robustness test, we excluded the alcohol variable and included *searches for tobacco* variable and found very similar results (see Table 5 in S1 Appendix). South Africa is one of the very few countries globally that have banned alcohol and tobacco sales during the Covid-19 pandemic. It is argued that these products contribute to the negative effects of the virus. The banning of these products had severe implications on different levels of society. Individuals see this as a major infringement of their human rights, negatively affecting their happiness. Furthermore, research done by Sommer et al. [38] proved that because of the presence of hordenine in beer, it significantly contributes to mood-elevation. In South Africa, which is well-known for its high per capita beer and alcohol consumption (Statistics South Africa [39]), also related to 'socialising', the ban on these products had a severe negative effect on happiness. Even in level 3, when the ban on alcohol sales was lifted, but still restricted, we found this negative relationship.

The restricted sale of alcohol and tobacco has indirect consequences for South Africans happiness via the economic impact since these industries are two of the largest industries in South Africa. They employ people across the whole supply chain from production to retail. Due to the ban on these industries, people can potentially lose their jobs. Lastly, the government sector forgoes all taxes on these products. This is against the backdrop of the recent downgrade of South Africa's debt rating to junk status in an already very uncertain fiscal environment. If all of these factors are aggregated, we can understand that the cumulative effect of the banning and restriction of sales of these products severely decreases the happiness levels. In Table 5 in S1 Appendix, we use tobacco searches instead of alcohol to estimate the determinants of happiness, which gives us results that are qualitatively similar to Table 3.

The number of tweets is negatively related to happiness (p-value<0.001). Previous research has shown that increases in the use of social media are negatively related to happiness (Rolland et al., Chae and Wilson et al. [34, 40, 41]). Noteworthy is that the number of tweets during the lockdown period increased significantly from an average of 60,708 per day before the lockdown to 80,000 per day after the lockdown indicating that more people used social media during the lockdown period.

Interesting is the result of the stay-at-home orders being positively related to happiness (p-value<0.001). On analysing the contents of the tweets, we find the following. South Africans are wary of contracting Covid-19, and therefore, they abide by the stay-at-home orders and

social distancing regulations to minimise the risk. That means that the stay-at-home orders in itself increase happiness; it is only once the other lockdown regulations are added that a total decrease in happiness levels are experienced.

Additional benefits revealed from analysing the tweets show that being at home provides a more peaceful and calmer environment compared to the rushed experience outside their homes. Also, people in the suburbs seem to be more convivial, with strangers greeting one another as people went for short walks around their neighbourhoods. In general, people have more time to spend with their loved ones. People earning salaries incur major savings, as there is less opportunity to spend money. People also save on commuting to and from workplaces and other destinations. One of the unexpected benefits of the stay-at-home orders is the much lower crime rates experienced in the country. Homes are constantly occupied, limiting the risk of residential robberies (-3.8 per cent). Other types of crimes such as murder (-72 per cent), rape (-87.2 per cent) and carjacking (-80.9 per cent) are much lower as well (Adapted from the speech of Police Minister Cele 2020 [42]).

In summary, what changed when the lockdown regulations to curb the spread of Covid-19 were implemented? It caused a significant decrease in happiness, and factors related to the lockdown regulations became relevant determinants of happiness.

## 5. Conclusions

In this paper, we used the Gross National Happiness Index (GNH), a real-time measure of well-being, derived from Big Data, to investigate whether lockdown regulations caused a decrease in happiness. Additionally, we determined which factors matter to happiness under these changed circumstances. We accomplished these aims by using two models: difference-in-difference and ordinary least squares.

We added to the current literature by determining *causality between lockdown and happiness and analysing the determinants of happiness after a lockdown in an extreme country case* using *real-time*, *Big Data*. Subsequently, this was the first study of its kind.

To determine if the decrease in GNH was due to the lockdown, specifically, we considered the estimated coefficient of the interaction variable 'lockdown and year'. We found a negative and statistically significant coefficient on the interaction variable, indicating that the lockdown caused a decline in GNH in 2020 compared to 2019.

As regards to the factors that are related to happiness after the lockdown was implemented, we found searches for alcohol (tobacco), the number of tweets and uncertainty about the future job market to be negatively related to happiness. In contrast, stay-at-home orders are positively related to happiness. Interesting is the negative and statistically significant squared relationship between new Covid-19 deaths and happiness, indicating that this relationship initially was positive but became negative over time.

Considering the results mentioned above, it ultimately means that if policymakers want to increase happiness levels and increase the probability to achieve the happiness levels of 2019, they must consider those factors that matter most to peoples' happiness. These factors include allowing creatures of habits some of their lost comforts by reinstating the sale of alcohol and tobacco. In saying that, we do advocate for responsible alcohol use by all South Africans and note that the significant effect of the ban on the sale of alcohol could be confounded by the restriction on social gatherings as well.

These results are important for countries in similar well-being situations, thus low levels of happiness, a diverse state of the economy and an increasing number of Covid-19 cases to evaluate what the effect of a strict lockdown is.

Additionally, policymakers should assure citizens that there is a credible plan to get the economy, which is currently in dire straits, back on track. Such an economic plan should

stimulate growth, create job opportunities and increase employment rates, supply the necessary infrastructure and deal with curbing vast budget deficits and debt burdens. Hopefully, such policies will fuel the dying embers of a dying economy and increase well-being levels.

Lastly, it would be remiss of us not to note the limitations of the study. First, we were restricted in the number of covariates we could add to our difference-in-difference model due to the limited observations and therefore potentially overidentifying the models. Second, regarding the inverted U-shaped relationship between new Covid-19 deaths and happiness, we acknowledge that there might be confounding factors at play, initially seen as 'positives' of the lockdown, but later turned into negatives. However, using alternative sets of covariates in the regression analyses, the inverted U-shape between new Covid-19 deaths and happiness remained. Therefore we trust that the revealed relationship is robust.

## Supporting information

**S1 Appendix. Robustness checks.**
(DOCX)

**S1 Data.**
(DTA)

## Acknowledgments

We would like to thank our colleagues Professor Emeritus John Knight from Oxford University and Dr Kelsey O'Connor from STATEC Luxembourg, for their generosity in providing feedback on the study.

## Author Contributions

**Conceptualization:** Talita Greyling, Stephanie Rossouw.

**Data curation:** Talita Greyling, Stephanie Rossouw, Tamanna Adhikari.

**Formal analysis:** Talita Greyling, Tamanna Adhikari.

**Investigation:** Talita Greyling, Stephanie Rossouw, Tamanna Adhikari.

**Methodology:** Talita Greyling, Stephanie Rossouw, Tamanna Adhikari.

**Project administration:** Talita Greyling, Stephanie Rossouw.

**Resources:** Talita Greyling.

**Software:** Talita Greyling, Stephanie Rossouw.

**Supervision:** Talita Greyling, Stephanie Rossouw.

**Validation:** Talita Greyling, Tamanna Adhikari.

**Visualization:** Talita Greyling, Tamanna Adhikari.

**Writing – original draft:** Talita Greyling, Stephanie Rossouw, Tamanna Adhikari.

**Writing – review & editing:** Talita Greyling, Stephanie Rossouw, Tamanna Adhikari.

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
