## [Decision Letter · Decision Letter 0]

21 Sep 2020

PONE-D-20-23334

The good, the bad and the ugly of lockdowns during Covid-19

PLOS ONE

Dear Dr. Tamanna Adhikari,

Thank you for submitting your manuscript to PLOS ONE. After careful consideration, we feel that it has merit but does not fully meet PLOS ONE’s publication criteria as it currently stands. Therefore, we invite you to submit a revised version of the manuscript that addresses the points raised during the review process.

We look forward to receiving your revised manuscript.

Kind regards,

Francesco Di Gennaro

Academic Editor

PLOS ONE

Journal Requirements:

2. Thank you for submitting the above manuscript to PLOS ONE. During our internal evaluation of the manuscript, we found significant text overlap between your submission and the following previously published works, some of which you are an author.

https://www.econstor.eu/bitstream/10419/217494/1/GLO-DP-0556.pdf

Please revise the manuscript to rephrase the duplicated text, cite your sources, and provide details as to how the current manuscript advances on previous work. Please note that further consideration is dependent on the submission of a manuscript that addresses these concerns about the overlap in text with published work.

We will carefully review your manuscript upon resubmission, so please ensure that your revision is thorough

"The funders had no role in study design, data collection and analysis, decision to

publish, or preparation of the manuscript."

Additional Editor Comments (if provided):

Dear Authors, follow reviewer suggestion you can improve your manuscript

Reviewers' comments:

Reviewer's Responses to Questions

**Comments to the Author**

1. Is the manuscript technically sound, and do the data support the conclusions?

Reviewer #1: No

Reviewer #2: Yes

Reviewer #3: Yes

2. Has the statistical analysis been performed appropriately and rigorously? 

Reviewer #1: I Don't Know

Reviewer #2: I Don't Know

Reviewer #3: Yes

3. Have the authors made all data underlying the findings in their manuscript fully available?

Reviewer #1: No

Reviewer #2: No

Reviewer #3: Yes

4. Is the manuscript presented in an intelligible fashion and written in standard English?

Reviewer #1: No

Reviewer #2: Yes

Reviewer #3: Yes

5. Review Comments to the Author

Reviewer #1: Thank you for giving me the opportunity to review your paper.

The paper attempted to validate the data source by comparing the gross national happiness index to other country circumstances. However, the happiness for a population can't measured by such a method! it should be based on scientific background.

The manuscript is mixed in most the sections. As the result should stands alone then the discussion comes to explain the results and link it with the earlier literature. Also, discussing the strengths or defending the limitation should lie under the discussion section not before.

Reviewer #2: This is a very interesting article covering an often overlooked factor in society - happiness. Using Twitter activity as a proxy, it seeks to assess the impact of the prevailing covid-19 pandemic lockdowns on happiness of the general population in a nation with pre-existing societal well-being challenges. It is thus both timely and pioneering in its approach as well as meet for the moment and useful for guiding public policy.

Further, the authors employ a unique dataset and did well in providing explanations for their analytic decisions. These would help inform the interpretations by the readers, post-publication, and add to the analysis pool in general.

1) Major Comments

a) The paper would be enriched if the graph visualising the 'inverted U-shape relationship' were added as this would provide some clarity about the variables on the axes. Moreover, the authors should comment on whether there might be other factors at play in/confounders to this relationship between happiness and COVID-19 deaths? The duration of the lockdown presumably aligns with the rising number of deaths and could perhaps impact happiness.

b) Figures should be placed properly in the body of the manuscript and not at the end.

c) Please include the units (e.g. New Daily Deaths per million) for each variable in Table 1.

d) When reporting results, could the authors please indicate the relevant p-values in the text and not just in the tables?

e) Could the authors please indicate the particular (qualitative) methodology adopted for the analysis of Tweets mentioned where the paper states "On analysing the contents of the tweets, we find the following. South Africans are wary...". Please state how many tweets were assessed and also clarify if it was the research team itself that did this analysis.

f) In saying "Also, people in the suburbs seem to be more colloquial [Should this read 'convivial'?], with strangers greeting one another.", could the authors mention if the strangers were meeting in person outside their homes during the lockdown or online?

g) Could the authors please include some limitations to the study at an appropriate position/in the conclusion section.

2) Minor Comments

a) In the introduction, the authors mention "what matters most"; a brief clarification of what it is that matters most would be helpful.

b) The authors also repeatedly make mention of 'Big Data'; a sentence or two explaining what constitutes Big Data would also be useful.

c) In section 2.1, could the authors please clarify what well-being in the South African context comprises? What are the indicators/dimensions they consider besides income?

d) In the first sentence of Section 2.3 'policy' should perhaps read 'policy-makers'.

Although the authors state that 'Yes - all data are fully available without restriction' and 'Data can be provided separately' I did not find any underlying data within the manuscript or as an addendum. Perhaps this might be due to the statement 'Tick here if your circumstances are not covered by the questions above and you need the journal’s help to make your data available.; Tick here if the URLs/accession numbers/DOIs will be available only after acceptance of the manuscript for publication so that we can ensure their inclusion before publication.' The reason notwithstanding, the absence of the underlying data makes me unable to comment on it in depth. I thus limit my comments to the data contained in the manuscript.

Reviewer #3: A very novel topic, with a very sound methodological analysis. Interesting findings; but I am concerned about the suggestions to reinstate the sale of alcohol and tobacco in order to increase happiness. Not only due to their impacts on public health, but also because the reported finding about them could be due to a confounding factor with social gatherings (where these items are usually consumed). I would suggest delving deeper about this in the discussion and I encourage you to align a message about responsible consumption of these substances.

6. PLOS authors have the option to publish the peer review history of their article (what does this mean?). If published, this will include your full peer review and any attached files.

Reviewer #1: **Yes: **Eman Almaghaslah MBBS MPH

Reviewer #2: No

Reviewer #3: **Yes: **eduardo alvarado

---

## [Author Response · Author response to Decision Letter 0]

6 Oct 2020

We thank the reviewers for the comments. Each of these has been comprehensively addressed, as set out in the response to the reviewers document (authors' response shown in italics). We have also made improvements throughout the paper to enhance its quality and contribution to the literature. Therefore, we believe the paper to be significantly improved as a result of the comments received from the reviewers. We have submitted a tracked changes version as well as a final manuscript version.

---

## [Decision Letter · Decision Letter 1]

2 Jan 2021

The good, the bad and the ugly of lockdowns during Covid-19

PONE-D-20-23334R1

Dear Dr. Rossouw

We’re pleased to inform you that your manuscript has been judged scientifically suitable for publication and will be formally accepted for publication once it meets all outstanding technical requirements.

Kind regards,

Francesco Di Gennaro

Academic Editor

PLOS ONE

Additional Editor Comments (optional):

dear authors congratulations

Reviewers' comments:

Reviewer's Responses to Questions

**Comments to the Author**

1. If the authors have adequately addressed your comments raised in a previous round of review and you feel that this manuscript is now acceptable for publication, you may indicate that here to bypass the “Comments to the Author” section, enter your conflict of interest statement in the “Confidential to Editor” section, and submit your "Accept" recommendation.

Reviewer #1: All comments have been addressed

Reviewer #2: All comments have been addressed

2. Is the manuscript technically sound, and do the data support the conclusions?

Reviewer #1: Yes

Reviewer #2: Yes

3. Has the statistical analysis been performed appropriately and rigorously? 

Reviewer #1: I Don't Know

Reviewer #2: Yes

4. Have the authors made all data underlying the findings in their manuscript fully available?

Reviewer #1: Yes

Reviewer #2: Yes

5. Is the manuscript presented in an intelligible fashion and written in standard English?

Reviewer #1: No

Reviewer #2: Yes

6. Review Comments to the Author

Reviewer #1: Thank you for considering all the comments. I read some economic styles and it is true it is different than the academic/ science-based style.

Please check for repetition, as some phrases were repeated more than two times.

Reviewer #2: Th authors have provided good responses to my comments and I do not object to the publication of the manuscript.

However, I wish to mention that the authors state that "All data has subsequently been uploaded to the correct PLOS ONE repository." They also state "ll relevant data are within the manuscript and its Supporting Information files."

As I was unable to find any special repository, again, I can only base my assessments on the data in the manuscript.

7. PLOS authors have the option to publish the peer review history of their article (what does this mean?). If published, this will include your full peer review and any attached files.

Reviewer #1: **Yes: **Eman Almaghaslah MBBS. MPH.

Reviewer #2: No

---

## [Editor Report · Acceptance letter]

7 Jan 2021

PONE-D-20-23334R1 

The good, the bad and the ugly of lockdowns during Covid-19 

Dear Dr. Rossouw:

I'm pleased to inform you that your manuscript has been deemed suitable for publication in PLOS ONE. Congratulations! Your manuscript is now with our production department. 

Kind regards, 

on behalf of

Dr. Francesco Di Gennaro 

Academic Editor

PLOS ONE